# Von Willebrand factor A1 domain stability and affinity for GPIbα are differentially regulated by its *O*-glycosylated N- and C-linker

**Klaus Bonazza[1], Roxana E Iacob[2], Nathan E Hudson[3], Jing Li[1], Chafen Lu[1], John R Engen[2], Timothy A Springer[1]\***

[1]Program in Cellular and Molecular Medicine, Boston Children's Hospital and Department of Biological Chemistry and Molecular Pharmacology, Harvard Medical School, Boston, United States; [2]Department of Chemistry and Chemical Biology, Northeastern University, Boston, United States; [3]Department of Physics, East Carolina University, Greenville, United States

**Abstract** Hemostasis in the arterial circulation is mediated by binding of the A1 domain of the ultralong protein von Willebrand factor (VWF) to GPIbα on platelets to form a platelet plug. A1 is activated by tensile force on VWF concatemers imparted by hydrodynamic drag force. The A1 core is protected from force-induced unfolding by a long-range disulfide that links cysteines near its N- and C-termini. The *O*-glycosylated linkers between A1 and its neighboring domains, which transmit tensile force to A1, are reported to regulate A1 activation for binding to GPIb, but the mechanism is controversial and incompletely defined. Here, we study how these linkers, and their polypeptide and *O*-glycan moieties, regulate A1 affinity by measuring affinity, kinetics, thermodynamics, hydrogen deuterium exchange (HDX), and unfolding by temperature and urea. The N-linker lowers A1 affinity 40-fold with a stronger contribution from its *O*-glycan than polypeptide moiety. The N-linker also decreases HDX in specific regions of A1 and increases thermal stability and the energy gap between its native state and an intermediate state, which is observed in urea-induced unfolding. The C-linker also decreases affinity of A1 for GPIbα, but in contrast to the N-linker, has no significant effect on HDX or A1 stability. Among different models for A1 activation, our data are consistent with the model that the intermediate state has high affinity for GPIbα, which is induced by tensile force physiologically and regulated allosterically by the N-linker.

**\*For correspondence:** springer@crystal.harvard.edu

**Competing interest:** The authors declare that no competing interests exist.

## Editor's evaluation

Through a series of elegant experiments, the authors concluded that the N-terminal linker together with its O-linked glycans stabilize the A1 in an inactive, native conformation. Removal of the N-linker or its O-linked glycans destabilizes the A1 structure and may activate the A1 into an intermediate, higher affinity state for GPIbα binding. The authors also concluded that the C-terminal linker and its O-linked glycans do not contribute to stabilizing the A1 structure, though they prevent GPIbα binding through steric hindrance and electrostatic repulsion. The conclusions are consistently supported by the data from multiple experimental techniques. The reported results are highly interesting for the hematology and biophysics communities.

## Introduction

The ultra-long length of the blood plasma protein von Willebrand factor (VWF) enables its activation by hydrodynamic forces at sites of hemorrhage (*Springer, 2014*; *Figure 1A, B*). VWF, a long concatemer of monomers linked head-to-head and tail-to-tail, undergoes a change of its multimeric superstructure from a random coil form (*Parker and Lollar, 2021*) to a thread-like, extended form when exposed to elongational flow (*Bonazza et al., 2015*; *Fu et al., 2017*). Hydrodynamic flow activates VWF to bind to glycoprotein Ib (GPIb) on platelets, the essential interaction required for hemostasis and thrombosis in the arteriolar circulation. Fluorescent imaging of single VWF concatemers tethered in a flow cell showed that binding of GPIbα not only requires elongation of VWF but also a half-maximal tensile force of ~20 pN transmitted through the backbone of VWF concatemers (*Fu et al., 2017*). Activation could be well fit to a model of transition from a low-affinity state 1 to a high-affinity state 2 (*Figure 1B*). Measuring A1–GPIbα binding and unbinding with laser tweezers also revealed two states, with a force dependent switch between two states with different on- and off-rates (*Kim et al., 2015*; *Kim et al., 2010*). In addition to such flex-bond measurements, A1 and GPIbα are also reported to form catch bonds, although these measurements lack single molecule fiduciary markers or fits to models from which on- and off-rates could be extracted (*Ju et al., 2013*). Despite multiple crystal structures of the A1 domain bound to GPIb, distinct conformational states of A1 that explain force-dependent regulation of A1 affinity for GPIbα have yet to emerge (*Blenner et al., 2014*; *Dumas et al., 2004*).

The force-responsive A1 domain is unusual among protein domains in being linked to neighboring domains through *O*-glycosylated, mucin-like segments (*Figure 1A*, blue hashed lines). Tensile force exerted on A1 is transmitted through these linkers. It has been suggested that A1-flanking sequences (*Auton et al., 2012*; *Interlandi et al., 2017*; *Ju et al., 2013*; *Miyata and Ruggeri, 1999*; *Sugimoto et al., 1993*) or domains (*Aponte-Santamaría et al., 2015*; *Ulrichts et al., 2006*) may shield the GPIbα-binding epitope and thereby inhibit otherwise tight binding to GPIb. It was also proposed that the N- and C-linkers interact with each other and form an autoinhibitory module that masks A1 (*Deng et al., 2017*). A force-dependent signature for breakage of this module has been reported that correlates

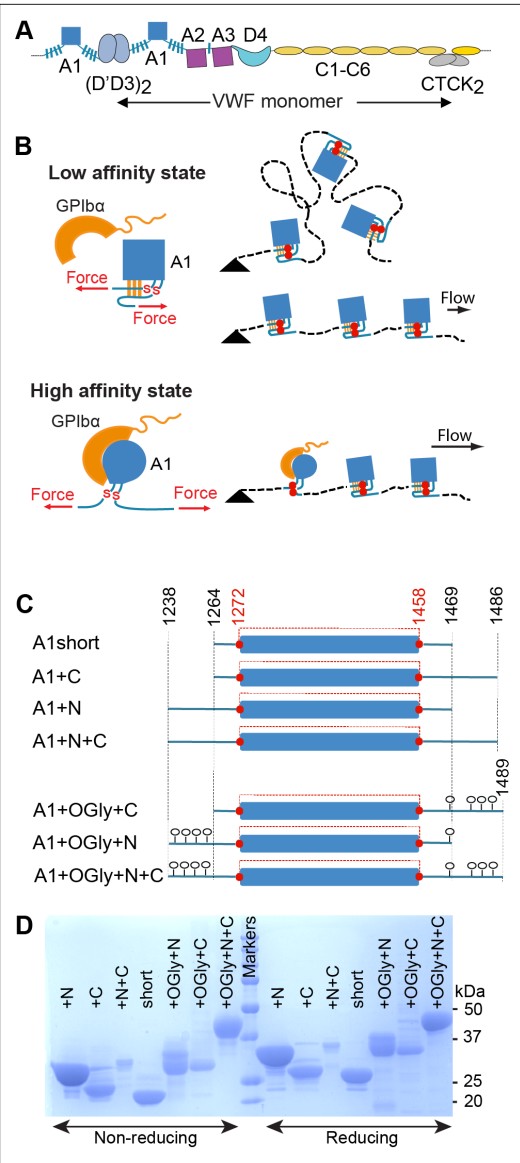

**Figure 1.** von Willebrand factor (VWF), hydrodynamic flow, A1 activation, and A1 linkers. (**A**) N- and C-terminal linkage and domain organization of a VWF monomer in a VWF concatemer. Monomers are connected head-to-head and tail-to-tail. (**B**) When exposed to elongational flow and tethered (black triangle) on a vessel wall, VWF concatemers extend to a linear shape. At higher elongational flow, which exerts higher mechanical tension along the length of the concatemer (highest at the tether point and zero at the downstream end), A1 undergoes transition from a low- to a high-affinity state as a consequence of tensile forces transmitted to it through its linkers. The disulfide bond between Cys1272 and Cys1458 is indicated with red SS. (**C**) A1 protein constructs studied here that differ in length of linkers and were expressed in *E. coli* (not glycosylated) or in mammalian cells (*O*-glycosylated) (O-glycans shown as lollipops). The long-range disulfide bond is schematized in red.

*Figure 1 continued on next page*

*Figure 1 continued*

(**D**) A1 constructs were subjected to sodium dodecyl–sulfate polyacrylamide gel electrophoresis (SDS–PAGE) and stained with Coomassie Blue. Amount of protein added per lane differed between protein constructs but was identical for each construct on reducing and nonreducing PAGE.

with the combined expected extension of the N- and C-linkers (*Arce et al., 2021*). An alternative mechanism has also been proposed by which the N-linker could regulate A1 affinity, that is, by regulating the relative stability of the A1 native and intermediate states. A1 has been found to have three states, native (N), intermediate (I), and denatured (D), while the homologous A2 and A3 domains in VWF have only two states, native and denatured (*Auton et al., 2007*). Lower stability of the A1 native state correlates with higher affinity (*Auton et al., 2012*; *Auton et al., 2010a*). This anticorrelation between A1 affinity and stability is also seen with von Willebrand disease type 2B gain-of-function mutations, which increase A1 affinity and decrease stability (*Auton et al., 2010b*; *Tischer et al., 2017*). Conversely, the G1342S decrease-in-function type 2M von Willebrand disease mutation decreases A1 affinity for GpIbα and increases A1 stability (*Auton et al., 2009*).

Mucin-like regions in proteins have unique characteristics distinct from both intrinsically disordered polypeptide segments and folded domains. The bulky *O*-linked glycans which include sialic acid that are attached to threonine and serine residues in mucins are highly solvated, repel one another, and make mucins extended, with an average length of ~2 Å/residue (*Clemetson, 1983*; *Woollett et al., 1985*). Electron microscopy showed that D3 and A1 in VWF, which are connected by the N-linker, have many possible orientations, and an interdomain distance as long as ~7 nm (*Zhou et al., 2011*). A1 is mechanically stabilized by a long-range disulfide bond between Cys1272 and Cys1458 (*Figure 1B, C*). Under tension, the core of A1 between these two cysteines, residues 1273–1457, is protected from complete unfolding. Only the portions of A1 external to this disulfide bond, residues 1238–1272 and 1458–1489, are directly subjected to the mechanical force transmitted through the spine of the VWF concatemer. However, multiple internal residues are noncovalently associated with folded external residues, and thus the stability of internal residues is expected to be affected by elongation of external residues by tensile force.

Here, to understand how binding of the VWF A1 domain to platelet GPIbα is regulated by its N- and C-terminal mucin-like linkers, we have measured the effects of both the glycan and polypeptide moieties of these linkers on stability, thermodynamics, and ligand-binding affinity and kinetics of A1. These linkers include the multiple serine and threonine residues that were found to be *O*-glycosylated during protein sequencing of native plasma VWF, including Thr-1248, Thr-1255, Thr-1256, and Ser-1263 in the N-linker (*Titani et al., 1986*). Mass spectrometry verified these assignments and showed a total of eight sialic acids over the four sites (*Solecka et al., 2016*). We show that both the N- and C-linkers affect affinity, that both their polypeptide and *O*-glycan moieties are important, and that the N-linker affects stability and hydrogen deuterium exchange (HDX) of A1 including in specific regions internal to its disulfide.

## Results

### Proteins

We compared the effect of the linkers N- and C-terminal to A1, which separate A1 from D3 and A2, respectively, and which contain *O*-glycosylation sites that have been chemically identified in native VWF (*Solecka et al., 2016*; *Titani et al., 1986*; *Figure 1*). Because the linkers are *O*-glycosylated in mammalian Expi293 cells and not in *E. coli*, we expressed A1 protein constructs in both cell types in order to test the effect in some assays of both polypeptide and *O*-glycan components of the linkers (*Figure 1C*). The N- and C-terminal boundaries of the constructs were based on allowing comparability to previous studies; references to the previous literature and a structural rationale for defining the minimal size of A1 as residues 1270–1463 and the maximal size of A1 plus linkers as residues 1240–1494 are described in Methods. To obtain A1 disulfide bond formation in bacteria, thioredoxin–A1 fusion proteins were expressed in *E. coli* SHuffle cells. The C-terminal boundary of the A1+N+C bacterial construct at Ser-1486 that was already on hand was extended by three residues in the mammalian A1+OGly+N+C construct to increase the likelihood of *O*-glycosylation of residue Ser-1486 and to include *O*-glycosylated residue Thr-1487 (*Solecka et al., 2016*; *Titani et al.,*

*1986*). Constructs were purified using His tag affinity and by Superdex S200 size exclusion, and for the bacterial constructs, also with an intervening heparin affinity chromatography step. As expected based on *O*-glycosylation, the mammalian constructs migrated more slowly and showed increased heterogeneity in sodium dodecyl–sulfate polyacrylamide gel electrophoresis (SDS–PAGE) compared to the corresponding bacterial A1 constructs (*Figure 1D*). Furthermore, faster migration of all seven A1 constructs in nonreducing than reducing SDS–PAGE confirmed that all contained the long-range disulfide bond.

## A1–GPIbα-binding kinetics and thermodynamics

Using biolayer interferometry (BLI), we measured A1 association to and dissociation from GPIbα immobilized through a biotinylated C-terminal avitag to BLI sensor tips. A1 preparations were subjected to S200 gel filtration the day of each measurement to remove aggregates. Global fits at all analyte concentrations to a single on- and off-rate for each A1 construct were good (*Figure 2*). In contrast, when gel filtration was omitted, data could not be fit to a single on- and off-rate. Differences among all seven A1 constructs in 150 mM NaCl showed that the linkers were of great importance in regulating A1 affinity for GPIbα (*Figure 2*). Comparisons to A1 short showed that nonglycosylated linkers in A1+N+C lowered affinity by 10-fold while the presence of glycosylated linkers in A1+OGly+N+C lowered affinity by 50-fold. The N-linker in A1+N lowered affinity by 4.3-fold while the C-linker in A1+C lowered affinity by 2.5-fold. Glycosylation of the N-linker was also more important than the C-linker; the affinity of A1+OGly+N was 10-fold lower than A1+N while the affinity of A1+OGly+C was only 2-fold lower than A1+C. *O*-Glycosylated linkers consistently lowered on-rates (*Figure 2B*). Overall, the results showed that both the polypeptide linker moiety and the *O*-glycan moiety of A1 linkers contributed to lowering A1 affinity for GPIbα, that the N-linker was more important than the C-linker in lowering affinity, and that the combined effect was very large at 50-fold.

To test the electrostatic contribution to binding affinity and on-rate, binding of A1 short and the three glycosylated constructs was additionally measured in 30 and 10 mM NaCl (*Figure 2*). Lowering NaCl concentration from 150 to 10 mM increased on-rates by an average of 17-fold for the four constructs. The effect of NaCl was predominantly on on-rate, since on- and off-rate together resulted in only a somewhat greater average increase in affinity of 23-fold (*Figure 2B*). Our results demonstrate that electrostatic complementarity between the binding interfaces in the basic A1 (pI 9.2) and acidic GPIbα (pI 5.5) domains (*Huizinga et al., 2002*) gives rise to electrostatic steering that speeds association kinetics.

To independently test affinity, and obtain binding thermodynamics, we used isothermal titration calorimetry (ITC). Although the GPIbα protein used in ITC lacked the avitag and its biotinylation, $K_D$ measurements in BLI and ITC were on average within 1.3-fold of one another and showed the same trends (*Figure 3*). A1 short showed a 39-fold increase in affinity compared to A1+OGly+N+C, the N-linker was more important than the C-linker, and the *O*-glycan moiety was more important than the polypeptide moiety in regulating affinity. Binding of all A1 protein constructs to GPIbα absorbed heats that showed the reaction was endothermic, that is, entropically driven (*Figure 3B*). The thermodynamics of A1–GPIbα binding showed enthalpy–entropy compensation (*Liu and Guo, 2001*; *Starikov and Nordén, 2007*): the higher the enthalpic contribution to binding, the higher the entropic cost, with all seven constructs following the same rank order for the enthalpic and entropic terms. Thus, the linkers affected both the enthalpic and entropic components of GPIbα binding.

## A1 stability and the thermodynamics of the intermediate state

The relative stabilities of A1 short and the three *O*-glycosylated A1 constructs to denaturation by heat or urea were measured by tryptophan fluorescence, whose emission maximum shifts to higher wavelength upon exposure to solvent during unfolding. The fluorescence intensity ratio (FIR), $I_{350}/I_{330}$, showed that constructs had three states, that is, exhibited two unfolding transitions (*Figure 4A, B, D*). Constructs containing the N-linker, that is A1+OGly+N+C and A1+OGly+N, were more stable to heat (Tm of 53.2 and 53.3°C for the first transition, respectively) than constructs lacking the N-linker, that is A1+OGly+C and A1 short (Tm of 49.9 and 50.7°C for the first transition, respectively) (*Figure 4A, B*). These results suggest that the N-linker stabilizes the native state of A1 relative to the intermediate state. Furthermore, all four constructs showed a second inflection point between 60° and 65° (second peak or shoulder of the derivative curve, *Figure 4B*). These results suggested two-step unfolding

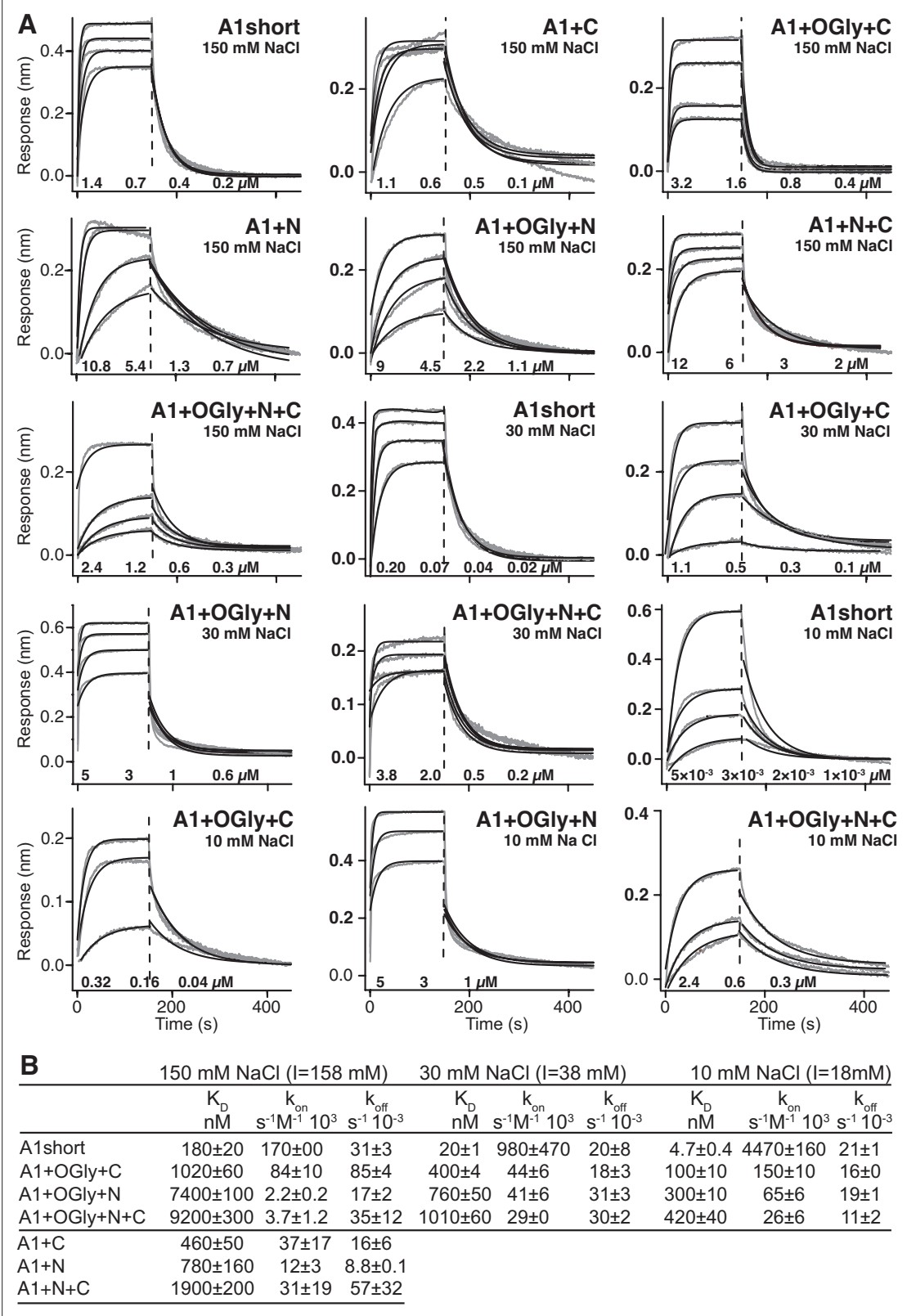

**Figure 2.** Effect of A1 linkers, their glycosylation, and ionic strength on GPIbα-binding kinetics. (**A**) Biolayer interferometry (BLI) traces in gray are shown for the indicated concentrations of the indicated A1 construct at the indicated NaCl concentration. In each panel, all A1 concentrations were fit globally to a one-site-binding model to obtain a single $k_{on}$ and $k_{off}$. Thin black lines show fits. Fitting errors ranged from 1% to 5% of the fit values. Vertical dashed lines mark the beginning of the dissociation phase. (**B**) Kinetic constants and derived $K_D$. Errors show difference from mean of $n = 2$ independent experiments.

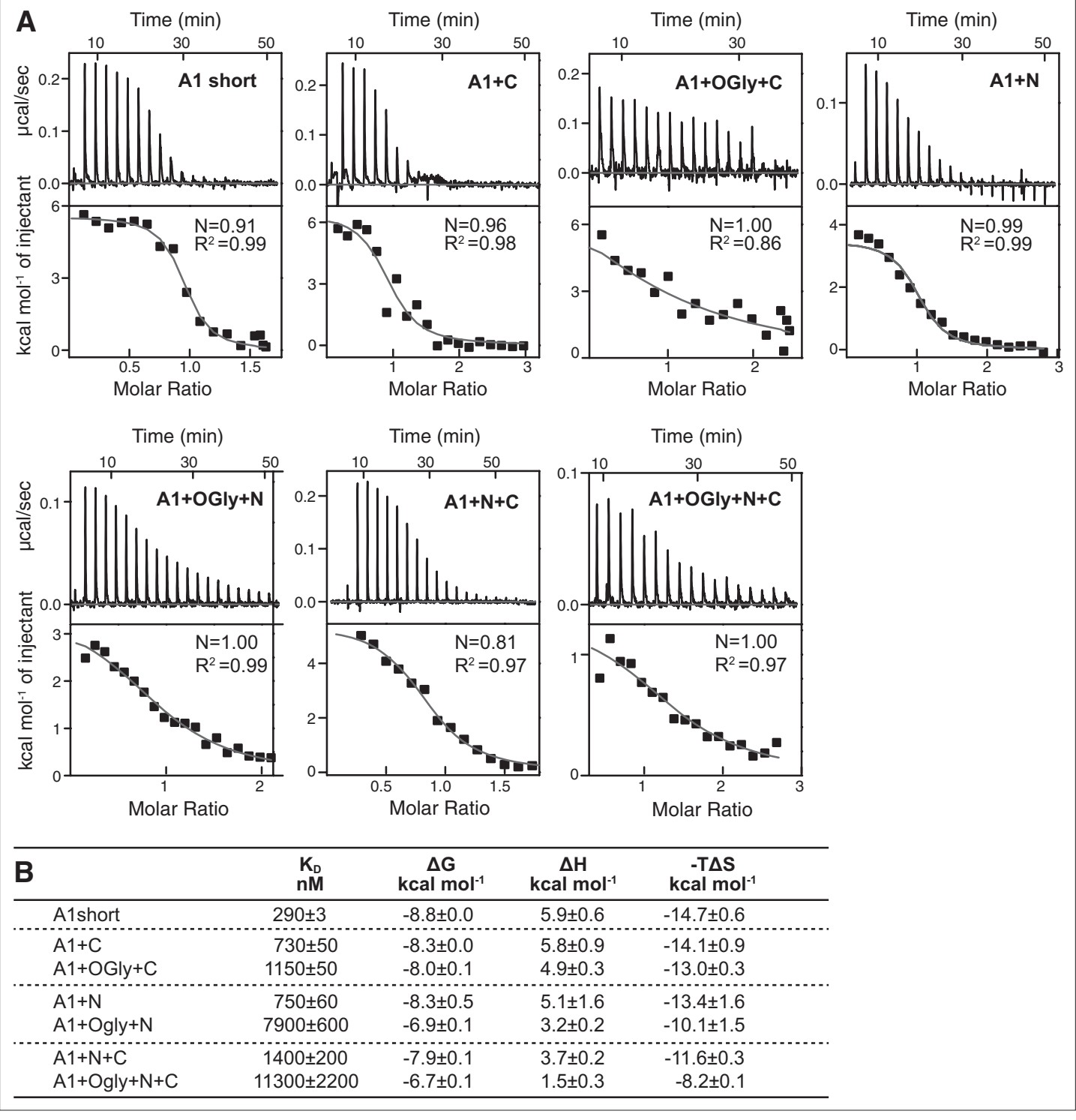

**Figure 3.** Effect of A1 linkers and their glycosylation on GPIbα-binding thermodynamics. (**A**) Panels show ITC traces of heat absorption after each injection (above) and fit of enthalpy to the binding isotherm (below) in 150 mM NaCl at 22°C. (**B**) Table summarizing the reaction Gibbs free energies ($\Delta G$) and the enthalpic ($\Delta H$) and entropic ($T\Delta S$) contributions. Errors show difference from mean of $n$ = 2 independent experiments, except for A1+N+C which shows the fitting error from one experiment.

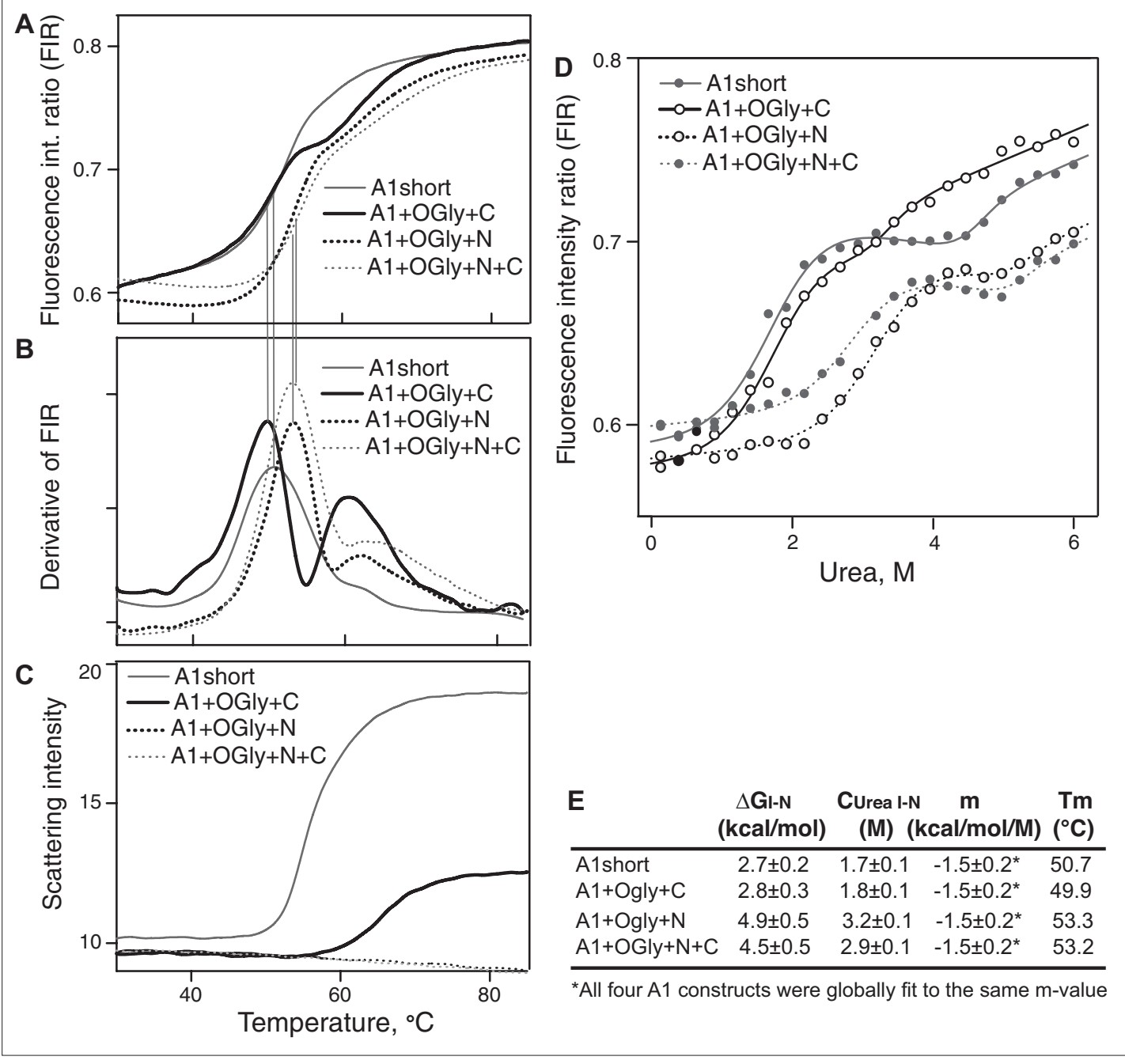

**Figure 4.** Stability of A1 constructs to heat and urea and free energy difference between the native and intermediate A1 states. (**A–C**) Effect of heating from 30 to 85°C on A1 stability and aggregation. (**A**) Unfolding was measured by excitation of tryptophan at 275 nm and measuring the ratio of the fluorescence intensity at 350 and 330 nm (fluorescence intensity ratio [FIR] = $I_{350}/I_{330}$). (**B**) The derivative of data in A (dFIR/d$T$) shows two unfolding transitions for each A1 construct and provides evidence for three states of A1: N, I, and U as described in Results. (**C**) Backscattered light intensity as a measure of A1 aggregation. (**D**) Unfolding of A1 by urea measured at 20°C by FIR as in panel A. Lines show fits to the three-state model of A1 unfolding as described in Methods and Results (see also Source Data). (**E**) Tabulation of the melting temperature of the native state (Tm) from panels A and B and the free energy difference between the native and intermediate states of A1 ($\Delta G_{I–N}$), the concentration of urea at which the I and N states are equally populated ($C_{\text{urea I–N}}$), and $m$, from the fits in panel D.

The online version of this article includes the following source data for figure 4:

**Source data 1.** A1 short.

**Source data 2.** A1 short + C.

**Source data 3.** A1 short + N.

**Source data 4.** A1 short + NC.

behavior and thus that A1 has three states, native (N), intermediate (I), and D (denatured) as previously observed (**Auton et al., 2007**).

A1 aggregation by heat was measured by backscattered light intensity. After the inflection point for the intermediate state, A1 short began to aggregate (**Figure 4C** compared to **Figure 4B**). A1+OGlyc+C required higher temperature for aggregation, which began to occur after the inflection point for the denatured state (**Figure 4C** compared to **Figure 4B**). Strikingly, the *O*-glycosylated N-linker completely protected A1+OGly+N and A1+OGly+N+C from aggregation at temperatures up to 85°C.

Denaturation in urea confirmed the existence of an intermediate state and was used to estimate the free energy of the intermediate state relative to the native state, that is, $\Delta G_{I-N}$. The A1 constructs showed two separate unfolding transitions, one between 1 and 4 M urea, and another above 5 M urea (**Figure 4D**). All constructs showed a plateau in between, or in the case of A1+OGly+C, followed a similar trend, but with a decrease in slope instead of a clear plateau. Similar three-state unfolding curves were previously seen for a construct similar to A1+N+C using circular dichroism (**Auton et al., 2007**). We fit the unfolding data to a three-state model. We assumed that all four constructs had the same *m*-value (a parameter that defines the cooperativity of transition) and that each state had a unique baseline FIR slope as a function of urea concentration that was shared among the four constructs (Methods). We thus determined the difference in energy between the I and N states ($\Delta G_{I-N}$), the concentration of urea at which the I and N states are equally populated ($C_{urea\ I-N}$), and *m*, a value that is related to the size of folded domains (**Ghosh and Dill, 2009**; **Santoro and Bolen, 1988**; **Figure 4E**). The results showed that the *O*-glycosylated N-linker markedly increased the stability of the native state and thus the gap in energy between the native and intermediate state, with higher $\Delta G_{I-N}$ values and transitions at higher urea concentrations for A1+OGly+N and A1+OGly+N+C than for A1 short and A1+OGly+C (**Figure 4E**).

## Changes in HDX dynamics among A1 constructs correlate with affinity and stability differences

HDX MS measures solvent accessibility and H-bonding and reports on the dynamics and stability of proteins (**Wales and Engen, 2006**). The *O*-glycosylated A1 constructs and A1 short were allowed to exchange their backbone amide hydrogens for deuterium in $D_2O$ for varying periods of time, digested with pepsin, and the kinetics of deuterium exchange were measured for 108 A1 peptides (91% coverage, 5.8× redundancy, **Figure 5** and **Figure 5—figure supplements 1–5**). HDX as a function of secondary structure and sequence position trended similarly for all four constructs studied (shown for A1 short in **Figure 5A** and for all in **Figure 5—figure supplements 1 and 2**). **Figure 5A** shows exchange for all 108 peptides at all time points, from 10 s to 4 hr. Over the A1 sequence from N- to C-terminal, the least exchange was seen for the β1-strand, the α1-helix, the β2- and β3-strands, the α3-helix, and the α6-helix, which all had at least one peptide with less than 20% deuteration at 4 hr. These slowly exchanging secondary structural elements neighbor one another and the GPIbα-binding site on A1 (**Figure 5D**).

Consistent differences in the amount of HDX were found in specific regions among the four A1 constructs (**Figure 5B, C** and **Figure 5—figure supplements 3–5**). The A1+OGly+N+C and A1+OGly+N constructs showed significantly less deuterium exchange (>0.7 Da) at almost all time points in peptides that were centered on the α1–β2 loop and the β3–α2 loop. These loops locate near to GPIbα (**Figure 5B–D**). We also observed reduced exchange in the α2–α3 loop, but the most meaningful difference was only seen at the 4 hr time point. The α1–β2 loop is nearby the long-range disulfide and in a region with multiple gain-of-function VWD type 2B mutations (**Figure 5D**). The β3–α2 loop is near to GPIbα and has four basic residues (labeled in **Figure 5D**) but they are not close enough to directly interact with acidic residues in the GPIbα leucine-rich repeats.

The HDX data showed no significant HDX differences between A1 short and A1+OGly+C. Further, the magnitude of HDX differences between A1 short and both A1+OGly+N+C and A1+OGly+N was very similar. However, while A1+OGly+N+C and A1+OGly+N were similar to one another in HDX, they both exchanged less deuterium than A1 short and A1+OGly+C, indicating that both A1 short and A1OGly+C were more dynamic than A1+OGly+N+C and A1+OGly+N. These HDX findings resembled the measurements of the stability of the native states relative to the intermediate states (**Figure 4**), which showed that A1+OGly+N+C and A1+OGly+N were similarly stable and more stable than A1 short and A1+OGly+ C, which were also similar in stability.

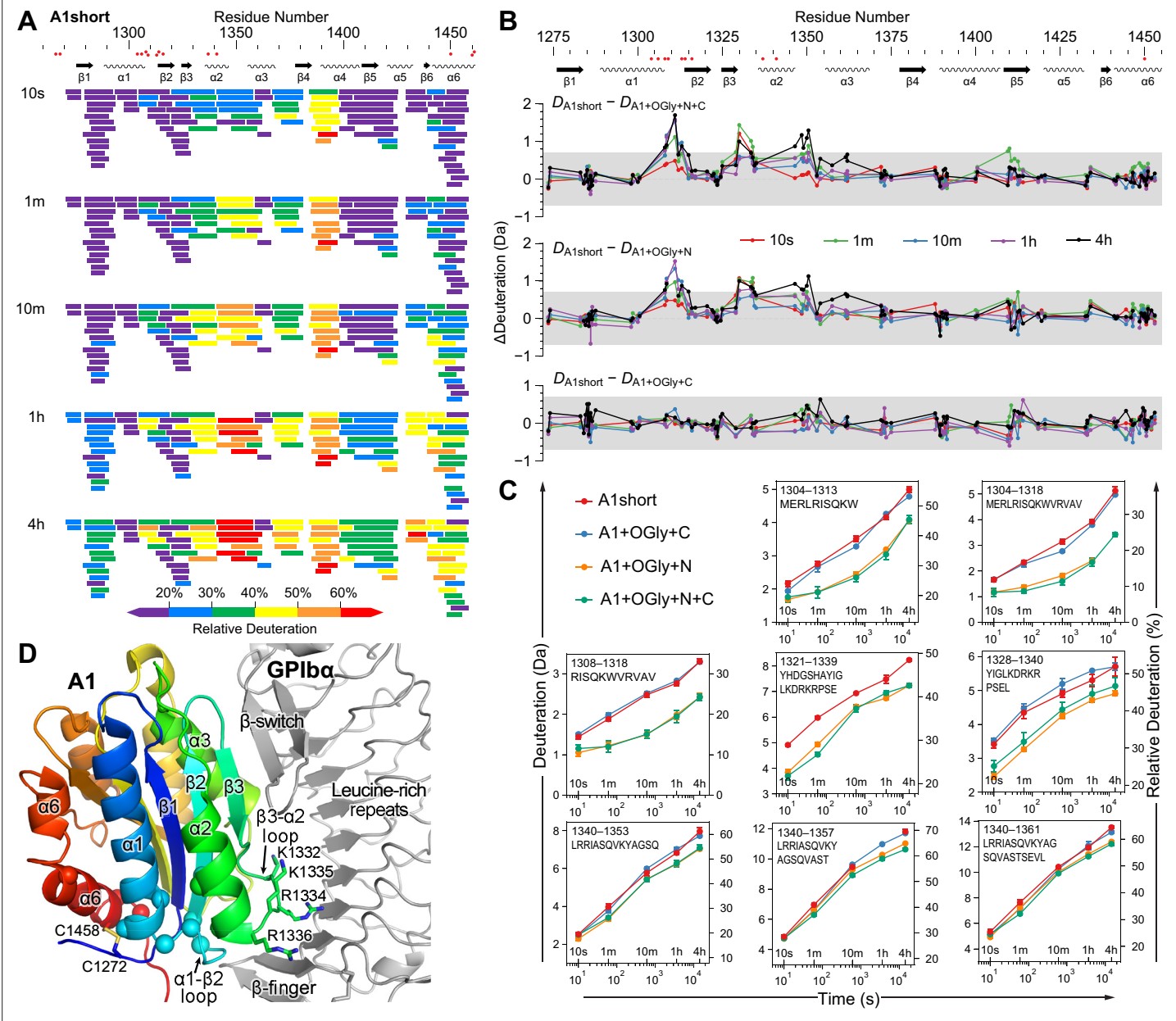

**Figure 5.** The N-linker decreases A1 dynamics measured by hydrogen deuterium exchange (HDX). (**A**) Relative deuterium exchange at all time points for A1 short as % of the available amide backbone H atoms in each peptide, colored according to the key. *Figure 5—figure supplements 1 and 2* show data for all constructs. (**B**) Difference (Δ) in HDX at all time points of A1 short minus HDX for the other three constructs as indicated. In (**A**) and (**B**), residues with VWD type 2B mutations are shown below sequence numbers as red circles. (**C**) Full HDX kinetics for selected peptides. *Figure 5—figure supplements 3–5* show data for all peptides. (**D**) Structure of A1 bound to GPIbα (PDB 1SQ0). A1 is colored rainbow from N- (blue) to C-terminus (red). Residues with VWD type 2B mutations are shown as Cα atom spheres. Labeled residues are shown in stick. GPIbα is shown in silver, from the β-finger to the β-switch.

The online version of this article includes the following figure supplement(s) for figure 5:

**Figure supplement 1.** Complete hydrogen deuterium exchange (HDX) peptide coverage at each indicated time point.

**Figure supplement 2.** Complete hydrogen deuterium exchange (HDX) peptide coverage at each indicated time point.

**Figure supplement 3.** Full hydrogen deuterium exchange (HDX) kinetics for peptides, one of three figures.

**Figure supplement 4.** Full hydrogen deuterium exchange (HDX) kinetics for peptides, two of three figures.

**Figure supplement 5.** Full hydrogen deuterium exchange (HDX) kinetics for peptides, three of three figures.

# Discussion

Among domains in proteins, the A1 domain of VWF is highly unusual in being separated from neighboring domains by mucin-like linkers and being activated by tensile force transmitted through those linkers that originates from hydrodynamic force on VWF concatemers and bound platelets. Here, we have investigated the effects of these linkers on A1 thermodynamics and function in binding to its ligand, GPIbα. Previous studies have examined the effect of the N-terminal linker or both linkers on various types of quantitative or nonquantitative binding assays and thermodynamics and found inhibitory effects on GPIbα binding from adding the N-linker or both linkers to A1 (*Arce et al., 2021*; *Auton et al., 2012*; *Deng et al., 2018*; *Deng et al., 2017*; *Interlandi et al., 2017*; *Ju et al., 2013*; *Miyata and Ruggeri, 1999*; *Nakayama et al., 2002*; *Tischer et al., 2017*; *Tischer et al., 2014*). We found that both the N- and C-linker decreased A1 affinity for GPIbα and that for both the N- and C-linker, *O*-glycosylated linkers decreased affinity more than polypeptide linkers. For the C-linker, both the *O*-glycan and polypeptide moieties were similar in importance, giving ~2.2-fold decreases in affinity each, while for the N-linker, the *O*-glycan component was more influential (9.5-fold) than the polypeptide alone (4.3-fold). While adding *O*-glycosylated linkers on the N- and C-terminal ends of A1 lowered affinity substantially (51- and 6-fold, respectively), the presence of both *O*-glycosylated linkers lowered binding little compared to the presence of the N-linker alone (1.2-fold). The separate contributions of the polypeptide and glycan moieties of the A1 N- and C-linkers had not previously been examined; however, *O*-glycosylation of both linkers together was reported to decrease affinity from 3.9 to 32 μM (*Tischer et al., 2020*). While these reported affinities differ from ours by two- to threefold, they agree in terms of the affinity difference caused by *O*-glycosylation. Kinetic measurements showed that the effect of both the polypeptide and *O*-glycan components of the linkers was primarily on-rate driven. Overall, among the seven constructs, on-rates varied by up to 80-fold (A1 short versus A1+OGly+N+C) while off-rates varied up to 10-fold (A1+OGly+C versus A1+N).

Rapid binding is especially important for proteins that function in hemostasis; electrostatics can help orient proteins for proper binding before they contact one another, as previously demonstrated using the ionic strength dependence of binding of soluble GPIbα to VWF tethered to the wall of a flow chamber (*Fu et al., 2017*). Our kinetic measurements in 10 mM compared to 150 mM NaCl showed a 20-fold increase in affinity for A1+OGly+N+C largely driven by a sevenfold increase in $k_{on}$, while VWF tethered to a flow chamber showed a 130-fold increase in affinity driven by an 11-fold increase in $k_{on}$.

In 150 mM NaCl, the high-affinity state of tethered, mechanically tensioned VWF had an affinity of 80 nM for GPIbα, compared to the $K_d$ values of 180 nM for A1 short and 9200 nM for A1+OGly+N+C measured here. Introduction of two VWD mutations into A1 was shown to increase affinity for GPIbα by 10-fold (*Blenner et al., 2014*). However, even though such a mutant would be expected to increase the affinity of the A1 short construct to a higher level than found for mechanically tensioned A1 in intact VWF, a crystal structure of A1 short with the two VWD mutations bound to GPIbα revealed only minor differences from a WT-GPIbα crystal structure (*Blenner et al., 2014*). Mechanical tension on the one hand and introduction of VWD mutations in the absence of tension and removal of the N- and C-linkers on the other hand may have different effects on the structure of the A1 domain, consistent with the hypothesis that A1 short with two VWD mutations bound to GPIbα did not reveal an A1 high-affinity state, despite an affinity for GPIbα comparable to that seen with mechanically tensioned VWF (*Blenner et al., 2014*).

We buttressed BLI affinity measurements here with ITC measurements of affinity and binding energies. While affinity measurements in the literature on essentially identical A1 fragments can differ up to 15-fold from those reported here (*Huizinga et al., 2002*), affinity measurements here on seven different A1 constructs by BLI and ITC were all within 1.6- to 1.04-fold of one another. We found it was necessary to remove aggregates from A1 preparations by gel filtration immediately prior to BLI measurements to avoid heterogeneous kinetics that required two on- and off-rates to fit, which appear in some of the literature on A1 binding to GPIbα (*Arce et al., 2021*; *Tischer et al., 2017*). In general, measurements in the liquid phase are more accurate than those in solid/liquid phases such as SPR and BLI, where a small amount of an avid aggregate can dominate binding kinetics.

We are unaware of previous thermodynamic measurements of A1 and GPIbα binding. ITC results show that the interaction of GPIbα with all seven A1 constructs was endothermic with a large entropic term and $\Delta G$ values that ranged from −8.2 to −14.7 kcal/mol at 22°C. Among all seven A1 constructs, the enthalpy and entropy terms were highly correlated, so that their rank orders were identical.

Entropy-driven binding of proteins is usually attributed to the increase in water entropy when waters are released from hydrophobic binding interfaces and is often counterbalanced by an increase in enthalpy (*Liu and Guo, 2001*; *Richards, 1977*; *Starikov and Nordén, 2007*).

Our HDX studies revealed that the *O*-glycosylated N-linker has a significant effect in stabilizing the α1–β2, β3–α2, and the α2–α3 loops, as shown by less deuterium incorporation. Previous HDX studies have shown that destabilizing A1 by reducing and alkylating its long-range disulfide, introducing VWD type 2B mutations, or omitting N-linker residues 1238–1260 increase HDX in the same regions (*Deng et al., 2018*; *Deng et al., 2017*; *Tischer et al., 2017*; *Tischer et al., 2014*). The finding that stands out from previous reports, which did not study the C-linker individually in HDX, is that in contrast to the N-linker, the C-linker did not significantly affect HDX in any of these regions.

We directly measured the stability of A1 by measuring its unfolding by heat and urea. We found evidence for three states by both techniques, consistent with a previous finding of an intermediate state that is hypothesized to represent a state with high affinity for GPIbα (*Auton et al., 2010a*; *Tischer et al., 2017*; *Tischer et al., 2014*). A plateau between the first and second transition clearly demonstrated the existence of the intermediate state for three of our four constructs, in contrast to the decrease in slope seen previously. This difference was likely due to measurement here of the fluorescence of the single tryptophan residue, Trp-1313 in A1, as opposed to circular dichroism, which measures contributions from all A1 secondary structures. Trp-1313 is in the α1–β2 loop that has HDX dynamics that are sensitive to the presence of the N-linker, locates near the GPIbα β-finger, is mutated in VWD type 2B, and although it may report denaturation of only a portion of A1, Trp-1313 discriminates between the native, intermediate, and denatured states of A1 similarly to circular dichroism.

Heat and urea denaturation had similar linker-dependent effects on stability of A1. A1+OGly+N and A1+OGly+N+C were each ~3°C more stable than A1 short and A1+OGly+C. Likewise, A1+OGly+N and A1+OGly+N+C each had native (N) states that were relatively more stable than their intermediate (I) states, by ~2 kcal/mol, than found for A1 short and A1+OGly+C. Additionally, the N-linker was sufficient to completely protect A1 against heat-induced aggregation up to 85°C. Since physiological tensile force on A1 may induce the I state, which could make A1 and with it an entire VWF concatemer susceptible to aggregation, one of the several important physiological roles of the N-linker and its *O*-glycosylation may be to protect VWF from aggregation. The neighboring A2 domain in VWF can be completely and reversibly unfolded by tensile force and can also be reversibly heated to 90°C without aggregation in the presence of its O-linked N-linker (the C-linker of A1) and its C-linker (*Xu and Springer, 2013*; *Zhang et al., 2009*), which may have a similar protective role physiologically to prevent aggregation during physiological partial (A1) and full (A2) unfolding.

Our results on the effects of the O-linked N- and C-linkers on affinity and stability differ. The C-linker decreased A1 affinity for GPIbα in two different measures, BLI and ITC, but had no effect on A1 stability as measured by HDX, melting temperature, and the relative stability of the A1 N and I states in urea denaturation experiments. These results suggest that the C-linker decreases affinity by direct mechanisms such as by steric or electrostatic repulsion. *O*-Glycans are decorated with sialic acid, which would repel negatively charged GPIbα. In contrast to the C-linker, the N-linker not only decreased affinity but also increased A1 stability as shown by significantly decreased HDX, increased melting temperature, and increased stability of the N state relative to the I state. The ~2 kcal/mol difference in ΔG between the N and I state for A1+OGly+N compared to A1 short corresponds to a ~30-fold difference in population of the I state. Thus, if the I state was the high-affinity state, much of the 40-fold difference in affinity between the I and N states could be accounted for by the shift in equilibrium toward the native state caused by the N-linker, and the remainder could be caused by steric and electrostatic repulsion of GPIbα. In VWF concatemers, the ΔG required for activation by mechanical tension of the high-affinity state was measured as 1.9 kcal/mol (*Fu et al., 2017*), considerably lower than the difference in energy between the I and N states measured for A1+OGly+N+C here of 4.7 kcal/mol. These differences suggest that the directional nature of energy input by mechanical tension allows for more efficient activation of A1 than urea denaturation and may induce a distinct and less disordered intermediate state.

Three different mechanisms have been proposed for how the linkers affect A1 binding to GPIbα. The proposal that the N- and C-linker associate with one another to form an autoinhibitory module (*Deng et al., 2018*; *Deng et al., 2017*) is not supported by our data that show that the N-linker alone is sufficient to inhibit binding of A1 to GPIbα. Furthermore, we find that the N- and C-linker are not

synergistic and to the contrary, have much less than additive effects on lowering binding affinity. It also has been proposed that residue Asp-1261 in the N-linker interacts with basic residues in the A1 domain to stabilize a low-affinity state of A1 (*Interlandi et al., 2017*). While our results do not bear on this mechanism, they do demonstrate that the O-linked glycans within the N-linker are sufficient to strongly inhibit binding of A1 to GPIbα. Finally, Auton et al. have described an intermediate state of A1 that is proposed to correspond to an activated high-affinity state of A1 (*Auton et al., 2007*; *Auton et al., 2012*; *Tischer et al., 2014*). Our data are consistent with and provide further evidence in support of this hypothesis.

The association of the N-linker but not the C-linker with A1 stability is consistent with structural analysis of A1 complexes with GPIbα that demonstrate that their association strains the N-terminal portion of A1 (reviewed in *Blenner et al., 2014*). The main site of GPIbα association is at the β-switch in its C-terminal cap (*Figure 5D*). The flexible β-finger in the N-terminal cap of GPIbα associates with A1 over a smaller region, including the α1–β2 loop, with few specific contacts. Furthermore, these contacts vary among GPIbα complexes. Strain in A1 upon binding GPIbα causes shifts in some residues in the α1–β2 loop and Cys-1272 away from GPIbα. Furthermore, the hydrogen bond between the backbone of Cys-1272 and the backbone of Arg-1308 (a residue often mutated in VWD), which is conserved in all structures of the isolated A1 domain, is lost in all A1–GPIbα complex structures. This loss admits a water molecule to the hydrophobic core of A1. These structural observations begin to define a pathway for conformational communication between the N-linker and regions of A1 that change upon GPIbα binding, including the α1–β2 loop, which showed increased HDX dynamics in the absence of the N-linker. Crystal structures of A1 diverge prior to residue 1270, suggesting that the O-glycosylated N-linker of A1 (defined in our experiments as residues 1238–1263) is natively disordered. It is now well established that natively disordered regions in proteins frequently have important roles in regulating protein allostery (*Motlagh et al., 2014*). We speculate that in the intermediate state, A1 would reshape to enable closer approach to the concave, acidic surface of GPIbα of more A1 basic residues, such as those in its β3–α2 loop shown in stick in *Figure 5D*.

In summary, our results show that the A1 N-linker, but not the C-linker, increases the stability of the A1 native state relative to the intermediate state, thus regulating conformational change. The previous paragraph discussed structural evidence for a pathway for communication from the N-linker to regions that are known to change structurally upon GPIbα binding and also have backbone motions that are influenced by the N-linker as shown by HDX. Additionally, our results suggest that the C-linker influences affinity for GPIbα by steric and possibly charge repulsion. The N-linker may repel GPIbα as well, in addition to regulating A1 conformation.

## Materials and methods
### Boundaries of A1 constructs

Boundaries for truncations of A1 used here were taken from what had been used in the literature and structural considerations. Constructs beginning here at residues 1238 and 1264 had previously been used in related papers in *Auton et al., 2007*; *Auton et al., 2012*; *Auton et al., 2010a*; *Auton et al., 2010b* and (*Blenner et al., 2014*), respectively. Similarly, a construct ending at residue 1469 was previously used in *Blenner et al., 2014*. Longer constructs used here ending at 1486 or 1489 ended at positions intermediate between longer constructs ending at 1480 (*Madabhushi et al., 2014*; *Tischer et al., 2020*) and 1493 (*Deng et al., 2018*). Twelve examples of A1 crystallized alone or in complexes (PDB codes 1m10, 1sq0, 1u0n, 1auq, 1u0o, 1ijk, 1ijb, 1fns, 1uex, 3hxo, 3hxq, and 4c2a) were superimposed to define boundaries between residues that were largely invariant in structure (part of the A1 domain) or variable among structures (part of linkers). Superimposition shows that the A1 backbone takes the same path beginning at residue F1270 and ending at residue E1463; the A1 disulfide links C1272 and C1458. The paths of two D3 assembly monomers (PDB code 6n29) are similar up to residue E1239 and the paths of five A2 monomers in PDB codes 3gxb and 3zqk are similar beginning at residues 1495. Thus, the minimum length of the A1 domain might be considered to be between residues 1270 and 1463 and the A1 linkers might be considered to extend up to the boundaries of the D3 and A2 domains, that is from residue 1240 to 1494. We are not aware of tests of the feasibility of obtaining good expression of constructs of these lengths and chose lengths that would allow comparison to previous results as described above.

## Glycosylated proteins

Human VWF A1 domain constructs as shown in *Figure 1C* (pre-pro-VWF amino acid residue numbering), beginning with residue 1238 or 1264 and ending with residue 1469 or 1489 and followed with a C-terminal 6xHis tag were cloned into a bicistronic IRES-GFP expression vector, ET8 (*Zhou and Springer, 2014*) and transfected into Expi293F cells using lipofectamine reagent (Invitrogen). Stable transfectants were selected in the presence of 500 μg/ml G418 (Geneticin) and subjected to two rounds of fluorescence-activated cell sorting. In each sort, the 8% most fluorescent cells were expanded in Expi293 expression medium. Culture supernatants were harvested after 5 days, supplemented with 20 mM HEPES (N-2-hydroxyethylpiperazine-N'-2-ethanesulfonic acid), pH 7.4 and 0.2 mM $NiCl_2$ (final concentration), and loaded onto a Ni-NTA agarose column by gravity (2 ml Ni-NTA agarose beads per 500 ml of culture supernatant). The column was pre-equilibrated in Washing Buffer (20 mM HEPES, pH 7.4, 150 mM NaCl, 0.2 mM $NiCl_2$). After loading the supernatant, the column was washed with 5 column volumes of Washing Buffer followed by 10 column volumes of 20 mM HEPES, 1 M NaCl, 16 mM imidazole, 0.2 mM $NiCl_2$. Protein was eluted with 20 mM HEPES, 150 mM NaCl, and 300 mM imidazole, and further purified by Superdex 200 (GE Healthcare) size-exclusion chromatography in 20 mM HEPES, 150 mM NaCl, pH 7.5.

Wild-type GPIbα, residues 1–290 of the mature protein with a C-terminal $His_6$ tag, with or without an intervening Avi-tag sequence (GLNDIFEAQKIEWHE), was purified from culture supernatant of HEK293 stable transfectants (*Blenner et al., 2014*) by Ni-NTA affinity chromatography and Superdex 200 size exclusion as above.

## Nonglycosylated proteins

The cDNA sequence encoding A1 beginning with residue 1238 or 1264 and ending with residue 1469 or 1486 (*Figure 1C*) was cloned into the pET32a vector with a thioredoxin fusion protein (*LaVallie et al., 1993*) and a TEV cleavage site at its N-terminus and a 6xHis tag at its C-terminus. Proteins were expressed in SHuffle cells (*Lobstein et al., 2012*), which were grown at 37°C in LB medium; at an A600 of 0.8, 1 mM IPTG was added and the temperature was shifted to 25°C. After 20 hr, cells were collected by centrifugation and lysed in a French press at 4°C in 20 mM HEPES (pH 7.4), 150 mM NaCl. The lysate was centrifuged at 17,000 × *g* at 4°C for 40 min. Ni-NTA agarose beads were added to the supernatant containing soluble A1 and shaken overnight at 4°C in 40 ml Falcon tubes (1.5 ml beads/40 ml supernatant). Beads were centrifuged at 2500 RPM for 5 min and resuspended in 10 ml buffer (20 mM HEPES, 150 mM NaCl, pH 7.4) per tube and vortexed for 2 min (first washing step). After two more washing steps, five cleaning steps were performed following the same procedure but with 20 mM HEPES, 1 M NaCl, and 16 mM imidazole. The beads were then packed into a column and eluted with 20 mM HEPES, 150 mM NaCl, 300 mM imidazole by gravity. The eluate was mixed with Tobacco Etch Virus (TEV) protease (A1:TEV mass ratio of 10:1) and dialysed against 20 mM HEPES, 150 mM NaCl overnight at 4°C. The thioredoxin-cleaved A1 was centrifuged for 10 min at 13,000 rpm to remove precipitate. The supernatant was further purified by heparin column (HiTrap Heparin HP, GE Healthcare), washed with 20 mM HEPES, 350 mM NaCl, and eluted with 20 mM HEPES, 800 mM NaCl. Finally, the protein was subjected to Superdex 200 size-exclusion chromatography as above.

## Storage and repurification

Protein concentrations were determined from their $A_{280}$ using the extinction coefficient calculated by ProtParam tool at the Expasy website. All proteins were stored at −80°C and subjected to a second round of size-exclusion chromatography on the day of measurements. This was essential to prevent protein aggregates from contributing to affinity and kinetics measurements, that is, to obtain data that can be reliably fitted with a 1:1 binding model.

## Kinetic binding measurements

The GPIbα used for kinetic measurements contained an N-terminal Avi-tag which was biotinylated using a BirA biotin-protein ligase kit (Cat #BirA500, Avidity, Aurora, CO). BLI used Octet RED384 instrument and software (ForteBio). Streptavidin-functionalized sensors were dipped in biotinylated GPIbα (1 μM) for 60 s and quenched with 10 μg/ml biotinyl-lysine for 40 s. Binding and dissociation responses were recorded at varying A1 concentrations in 20 mM HEPES, pH 7.5, 0.02% Tween-20, and either 10, 30, or 150 mM NaCl as indicated. Response in a buffer-only reference well was subtracted

(internal reference). The experiment was also carried out using the same concentrations of A1 and the internal reference but with sensors that had no absorbed GPIbα (parallel reference). Parallel reference responses were subtracted from the experimental responses with Octet software. Data at different A1 concentrations were fit globally to single $k_{on}$ and $k_{off}$ values for each experimental condition and plotted using Originlab (Origin, Northampton, MA). The ionic strength of 20 mM HEPES, pH 7.5 was calculated using https://www.liverpool.ac.uk/pfg/Research/Tools/BuffferCalc/Buffer.html.

## Isothermal calorimetry

Proteins were dialyzed overnight against 150 mM NaCl, 20 mM Tris–HCl, pH 7.5, degassed, and centrifuged at 20,000 × g for 10 min. GPIbα (250 µM except 550 µM with A1+OGly+N+C) was titrated into an A1 protein solution (22 µM except 50 µM with A1+OGly+N+C) in a MicroCal iTC200 (GE Healthcare Life Sciences). A priming injection of 0.4 µl (not included in data analysis) was followed by 2 µl injections every 180 s. Data averaged over 2 s windows were analyzed using Originlab 7.

## Stability to denaturant and heat

For urea denaturation, A1 (0.25–0.5 mg/ml) in 10 µl of 20 mM HEPES, pH 7.5, 150 mM NaCl was mixed with 10 µl of 200 mM HEPES, pH 7.5, 1.5 M NaCl, ×µl of 7.5 M urea, and (80-x) µl of water, where x gave final urea concentrations of 0.135–6 M in 24 concentration steps in 100 µl, and equilibrated for 24 hr at 20°C. Intrinsic Trp fluorescence was measured at 330 and 350 nm upon excitation at 275 nm with a Prometheus NT48 (NanoTemper) with automatic adjustment of excitation power.

FIR, emission intensities at 350 nm over 330 nm, was fit as a function of urea concentration to a three-state transition model using the linear extrapolation method (*Santoro and Bolen, 1988*). *FIR* at each urea concentration ($C_{urea}$) is modeled as the population weighted contribution from the native state (N), intermediate state (I), and the denatured state (D): $FIR\ (C_{urea}) = FIR_N * P_N + FIR_I * P_I + FIR_D * P_D$ where $FIR_N$, $FIR_I$, and $FIR_D$ represent the linear baseline for the N, I, and D states, respectively, which can be expressed as $FIR_j = b_j + s_j \cdot C_{Urea}$, with $b_j$ and $s_j$ as intercept and slope of the $j$ state. $P_N$, $P_I$, and $P_D$ represent the population of N, I, and D states at each urea concentration, respectively. The population of state $j$ ($P_j$) in the system can be related to the free energy of each state (N, I, D) based on the Boltzmann distribution, $P_j = \frac{\text{Exp}\left(\frac{-\triangle G_j}{\text{R}\cdot T}\right)}{\text{Exp}(\frac{-\triangle G_N}{\text{R}\cdot T}) + \text{Exp}(\frac{-\triangle G_I}{\text{R}\cdot T}) + \text{Exp}(\frac{-\triangle G_D}{\text{R}\cdot T})}$, where $R$ is the gas constant, and $T$ is absolute temperature of the system. With the N state as reference state ($\Delta G_N = 0$), the free energy of the I and D states at each urea concentration can be expressed as $\Delta G_I(C_{Urea}) = \Delta G_I^0 + m_I \cdot C_{Urea}$ and $\Delta G_D(C_{Urea}) = \Delta G_D^0 + m_D \cdot C_{Urea}$, respectively, where $\Delta G_I^0$ and $\Delta G_U^0$ are the free energy of the I and D state in absence of urea, and $m_I$ and $m_D$ are the cooperativity of transition from the N to I state, and from N to D state, respectively. For a cooperative two-state transition induced by a particular denaturant, the $m$-value is proportional to the surface area exposed upon protein denaturation and is generally proportional to the protein size. The linear relationship between $m$-values and globular protein chain lengths have been reported for denaturant-induced protein unfolding (*Ghosh and Dill, 2009*). Thus, the final fitting function for the three-state transition is as follows, where $b$ and $s$ are baseline intercept and slope values, respectively, for N or I states:

$$FIR(C_{urea}) = \frac{(b_N + s_N \cdot C_{Urea}) + (b_I + s_I \cdot C_{Urea}) \cdot \text{Exp}(-\frac{\Delta G_I^0 + m_I \cdot C_{Urea}}{R \cdot T}) + (b_U + s_U \cdot C_{Urea}) \cdot \text{Exp}(-\frac{\Delta G_D^0 + m_D \cdot C_{Urea}}{R \cdot T})}{1 + \text{Exp}(-\frac{\Delta G_I^0 + m_I \cdot C_{Urea}}{R \cdot T}) + \text{Exp}(-\frac{\Delta G_D^0 + m_D \cdot C_{Urea}}{R \cdot T})} \quad (1)$$

Nonlinear least square fit to *Equation 1* was employed to fit the urea denaturation data for four A1 constructs. The data were globally fit with shared $s_N$, $s_I$, $s_D$, $m_I$, and $m_D$, and individual $b_N$, $b_I$, $b_D$, $\Delta G_I^0$, and $\Delta G_D^0$ for each construct. The rationale for fitting the baseline slope for each state to a shared value between the four constructs is that the N- and C-terminal linker does not contain any tryptophan. The rationale for the four constructs to share the same $m$-values for the I and D states is that the four constructs only differ in the O-glycosylated linkers which are not folded.

For heat denaturation, A1 (0.25–0.5 mg/ml) was in 20 mM HEPES, pH 7.5, 150 mM NaCl. FIR was measured in the Prometheus NT48 while temperature was ramped from 30 to 85°C at a rate of 1 °C/min using an excitation power of 50%. Additionally, the intensity of backscattered light was recorded as a measure of aggregation.

## HDX mass spectrometry

HDX experiments were essentially as reported previously (*Iacob et al., 2013*; *Le et al., 2018*). Comprehensive experimental details and parameters are provided in *Table 1*, in the recommended (*Masson et al., 2019*) tabular format. All HDX MS data have been deposited to the ProteomeXchange Consortium via the PRIDE (*Perez-Riverol et al., 2019*) partner repository with dataset identifier PXD029942.

Briefly, A1 short and each *O*-glycosylated protein construct (28 µM in 20 mM HEPES, 150 mM NaCl) was diluted 15-fold into 20 mM HEPES, 150 mM NaCl, 99% $D_2O$ (pD 7.5) at room temperature. At deuterium exchange time points from 10 s to 240 min, an aliquot was quenched by adjusting the pH to 2.5 with an equal volume of 300 mM sodium phosphate, 0.25 M tris (2-carboxyethyl)phosphine hydrochloride (TCEP-HCl), $H_2O$. Samples were analyzed as previously described (*Iacob et al., 2013*; *Wales et al., 2008*). They were digested offline with 10 mg/ml pepsin in water for 5 min on ice, then injected into a custom Waters nanoACQUITY UPLC HDX Manager. All mass spectra were acquired using a Waters Synapt G2-Si HDMS[E] mass spectrometer. Comparison experiments were done under identical experimental conditions such that deuterium levels were not corrected for back exchange and are therefore reported as relative (*Wales and Engen, 2006*). All experiments were performed in duplicate. The error of measuring the mass of each peptide was ±0.15 Da in this experimental setup. The peptides were identified using PLGS 3.0.1 software and the HDX MS data were processed using DynamX 3.0 (Waters Corp., USA). Peptides common to A1+OGly+N+C and the shorter constructs were followed with HDX, with an overall sequence coverage of 90.9%. In total, 108 peptic peptides were followed with HDX uptake plots (*Figure 5—figure supplements 1–5*).

**Table 1.** Hydrogen deuterium exchange (HDX) MS data summary and list of experimental parameters.

| Dataset | A1 short | A1N | A1C | A1NC |
|---|---|---|---|---|
| HDX reaction details* | 15-Fold dilution with labeling buffer at 20°C, final $D_2O$ level = 93.3%, $pH_{read}$ = 7.10; equal volume quench buffer at 0°C, $pH_{read}$ = 2.5 | | | |
| HDX time course | 10 s, 1 m, 10 m, 1 hr, 4 hr | | | |
| HDX controls | 3 undeuterated for each condition | | | |
| Back exchange | 30–35% | | | |
| Number of peptides | 102 | 103 | 101 | 108 |
| Sequence coverage | 93.2% | 92.5% | 91.8% | 90.9% |
| Avg. peptide length (a.a.) | 12.3 | 12.2 | 12.3 | 12.3 |
| Redundancy | 5.4 | 5.14 | 4.89 | 5.77 |
| Replicates | 2 technical for each condition | | | |
| HDX repeatability† | ±0.15 relative Da | | | |

*Labeling buffer: 20 mM HEPES, 150 mM NaCl, 99% $D_2O$, pD 7.5; quench buffer: 300 mM sodium phosphate, 0.25 M Tris (2-carboxyethyl)phosphine hydrochloride (TCEP-HCl), $H_2O$ pH 2.5.

†All reported values are the average relative deuterium level as given by the DynamX software, which in some cases is an average of more than one charge state, across all peptides in both technical replicates. No statistical tests were applied to the HDX MS measurements.

## Acknowledgements

We would like to thank Prof. Thomas Wales for helpful discussions and a research collaboration with the Waters Corporation (JRE). We thank Sha Wang and Amy Xu for early work. We thank Yang Su for writing and using code to display HDX MS data for figure preparation. We acknowledge support from the National Hemophilia Foundation and from the Austrian Science Fund (FWF), Erwin Schrödinger Fellowship number J4081-B21, NIH grant F32HL126386, and NIH grant R01-HL148755.

## Additional information

### Funding

| Funder | Grant reference number | Author |
| --- | --- | --- |
| National Hemophilia Foundation | Postdoctoral fellowship grant | Klaus Bonazza |
| Austrian Science Fund | Erwin Schrödinger Fellowship J4081-B21 | Klaus Bonazza |
| National Institutes of Health | F32HL126386 | Nathan E Hudson |
| National Institutes of Health | R01-HL148755 | Timothy A Springer |

The funders had no role in study design, data collection, and interpretation, or the decision to submit the work for publication.

### Author contributions

Klaus Bonazza, Formal analysis, Funding acquisition, Investigation, Writing – original draft, Writing – review and editing; Roxana E Iacob, Formal analysis, Investigation; Nathan E Hudson, Funding acquisition, Investigation; Jing Li, Formal analysis, Writing – review and editing; Chafen Lu, Methodology, Writing – review and editing; John R Engen, Formal analysis, Funding acquisition, Methodology, Writing – review and editing; Timothy A Springer, Conceptualization, Funding acquisition, Methodology, Supervision, Writing – original draft, Writing – review and editing

### Author ORCIDs

John R Engen ![ORCID] http://orcid.org/0000-0002-6918-9476
Timothy A Springer ![ORCID] http://orcid.org/0000-0001-6627-2904

### Decision letter and Author response

Decision letter https://doi.org/10.7554/eLife.75760.sa1
Author response https://doi.org/10.7554/eLife.75760.sa2

## Additional files

### Supplementary files

• Transparent reporting form

### Data availability

Figure 4—source data 1–4 contain the numerical data used to generate the figures. All HDX MS data have been deposited to the ProteomeXchange Consortium via the PRIDE (Perez-Riverol et al., 2019) partner repository with dataset identifier PXD029942.

The following dataset was generated:

| Author(s) | Year | Dataset title | Dataset URL | Database and Identifier |
|---|---|---|---|---|
| Bonazza K | 2021 | ProteomeXchange dataset PXD029942 | http://proteomecentral.proteomexchange.org/cgi/GetDataset?ID=PXD029942 | ProteomeXchange, PXD029942 |

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
