## [Editor Report]

Through a series of elegant experiments, the authors concluded that the N-terminal linker together with its O-linked glycans stabilize the A1 in an inactive, native conformation. Removal of the N-linker or its O-linked glycans destabilizes the A1 structure and may activate the A1 into an intermediate, higher affinity state for GPIbα binding. The authors also concluded that the C-terminal linker and its O-linked glycans do not contribute to stabilizing the A1 structure, though they prevent GPIbα binding through steric hindrance and electrostatic repulsion. The conclusions are consistently supported by the data from multiple experimental techniques. The reported results are highly interesting for the hematology and biophysics communities.

---

## [Decision Letter]

**Decision letter after peer review:**

Thank you for submitting your article "Von Willebrand factor A1 domain affinity for GPIbα and stability are differentially regulated by its O-glycosylated N-linker and C-linker" for consideration by *eLife*. Your article has been reviewed by 3 peer reviewers, and the evaluation has been overseen by a Reviewing Editor and Mone Zaidi as the Senior Editor. The following individual involved in review of your submission has agreed to reveal their identity: P. Clint Spiegel (Reviewer #2).

Essential revisions:

In this paper, the authors use several biophysical tools, including biolayer interferometry, isothermal calorimetry, protein denaturation, and hydrogen-deuterium exchange mass spectrometry, to systemically dissect the influence of the N- and C-terminal polypeptide linkers and O-glycan moieties of the VWF A1 domain on A1's activation state and binding affinity to platelet receptor GPIbα.

Through a series of elegant experiments, the authors concluded that the N-terminal linker together with its O-linked glycans stabilize the A1 in an inactive, native conformation. Removal of the N-linker or its O-linked glycans destabilizes the A1 structure and may activate the A1 into an intermediate, higher affinity state for GPIbα binding. The authors also concluded that the C-terminal linker and its O-linked glycans do not contribute to stabilizing the A1 structure, though they prevent GPIbα binding through steric hindrance and electrostatic repulsion. The data and the figure presented in the manuscript are overall clear, and the conclusions are consistently supported by the data from multiple experimental techniques. The reported results are highly interesting for the hematology and biophysics communities.

1) Define the A1 domain in terms of the sequence that delimits it and the sequences that define the N- and C- linkers and provide justification for the selection.

2) Show the standard deviation of the fit instead of showing the average of two experiments in figure 2 and provide additional description for the methods used to purify protein constructs in the gel images as the bands appear differentially pure, in different amounts in figure 1.

*Reviewer #1 (Recommendations for the authors):*

In this paper, the authors use several biophysical tools, including biolayer interferometry, isothermal calorimetry, protein denaturation, and hydrogen-deuterium exchange mass spectrometry, to systemically dissect the influence of the N- and C-terminal polypeptide linkers and O-glycan moieties of the VWF A1 domain on A1's activation state and binding affinity to platelet receptor GPIbα.

Through a series of elegant experiments, the authors concluded that the N-terminal linker together with its O-linked glycans stabilize the A1 in an inactive, native conformation. Removal of the N-linker or its O-linked glycans destabilizes the A1 structure and may activate the A1 into an intermediate, higher affinity state for GPIbα binding. The authors also concluded that the C-terminal linker and its O-linked glycans do not contribute to stabilizing the A1 structure, though they prevent GPIbα binding through steric hindrance and electrostatic repulsion. The data and the figure presented in the manuscript are overall clear, and the conclusions are consistently supported by the data from multiple experimental techniques. The reported results are highly interesting for the hematology and biophysics communities

I have only a few suggestions for revision and improvement, detailed below.

1) The length of the long C-linker is inconsistent among the constructs. While the constructs expressed from *E. coli* end at residue 1486, the mammalian constructs end at residue 1489. No further justification was given by the authors. Moreover, since the A2 domain starts at residue 1494, why didn't the authors include residues 1490-1493?

2) In regard to Figure 2, some of the global fits are not of high quality (e.g., A1+C with 150 mM NaCl). The authors should show the standard deviation of the fit instead of showing the average of two experiments.

3) It would be worth discussing the difference between the intermediate state identified by the thermal/chemical denaturation experiments and the more physiological mechanically tensioned structure. The intermediate state may or may not be the same high-affinity conformation induced by tensile pulling, since heat/urea destabilizes the entire A1 structure and tensile force/tension is not expected to spread into the structures protected by the long-ranged disulfide.

4. Lines 223-224: It would be nice to specify the exact constructs and conditions that account for the 80-fold and 10-fold differences.

*Reviewer #2 (Recommendations for the authors):*

Bonazza, Klaus et al. investigate the molecular details for how von Willebrand factor (VWF) binds to Glycoprotein Ib (GPIb) on platelets. Previous studies have demonstrated that VWF binds to GPIb only under conditions of high tensile force through an unfolding event that occurs for the A1 domain of VWF. This study emphasizes the role(s) of the unstructured linkers, along with their associated glycosylation states, that are N- and C-terminal to the A1 domain. Their data demonstrate that both linkers decrease the affinity of VWF A1 for GPIb; the N-terminal linker decreases the affinity more strongly whilst stabilizing the A1 fold, and glycosylation of the linkers contributes more to the decreased affinity than the peptide component. With biolayer interferometry (BLI), they present association and dissociation curves for this intermolecular interaction that fit well to a one-site binding model to calculate on- and off-rates with high confidence, showing a decrease in affinity for various constructs containing the N- and/or C- terminal linkers (with and without glycosylation). Additionally, they use BLI to show that the interaction is largely electrostatic by calculating higher affinities at lower ionic strength conditions. They further support the BLI data with measuring equilibrium binding thermodynamics with isothermal titration calorimetry (ITC). This experimental approach allows them to demonstrate similar affinity values to the BLI results, but also provides a deeper understanding for the thermodynamic basis for the VWF/GPIb interaction; the binding event is endothermic (entropically driven). In order to investigate how the N- and C-linkers affect protein stability, they employed both thermal and chemical denaturation measured with intrinsic tryptophan fluorescence. These data suggest that the A1 domain adopts an intermediate, partially unfolded state, which has differential stability dependent on the presence of the linkers; the N-terminal linker stabilizes the 'native' state, thus restricting the partially unfolded state that is hypothesized to be the GPIb binding state. Following back scattering intensity as a function of temperature in the same experiment, they also show that the N-linker prevents aggregation of the A1 domain. Lastly, the authors employ H/D exchange measured by mass spectrometry (HDX-MS) to show that the presence of the N-linker specifically increases protection patterns (decreased H/D exchange) for regions of the A1 domain that are adjacent to the GPIb binding interface and the internal disulfide bond. Taken together, what emerges from this study is a model for VWF A1/GPIb association that is regulated by glycosylated linkers whereby the N-linker stabilizes a 'native' conformation, which is only perturbed to an intermediate, partially unfolded state under high tensile force, leading to the activation of a high affinity binding state for GPIb. The data presented in this study are clearly communicated and appropriately controlled and interpreted.

Some perceived weaknesses are as follows:

1. The authors appropriately report a gel image of their purified protein constructs. The bands on the gel appear differentially pure, in different amounts, and there is no description in the manuscript about the relative 'diffuse' nature of the bands for glycosylated protein constructs.

2. The study describes the rigorous treatment of their thermal and chemical denaturation curves, and they briefly address the difference between measuring protein unfolding via intrinsic tryptophan fluorescence vs. circular dichroism. None of the curves presented illustrate error bars, but they do provide a measure of error in the associated table. Their measurements in this experiment are exclusively dependent on the environment of a single tryptophan residue in the A1 domain structure; their interpretations of the unfolding curves are warranted, but not without the caveat that it is only measuring the local environment of the tryptophan and not the entire protein structure.

My recommendations for the authors concern the gel image in figure 1 and the Materials and methods section. For figure 1, the protein bands are underwhelming. The reader would have more confidence in the study if this gel demonstrated a more uniform concentration of the protein constructs used in the study. Some of the bands are quite large and dominate the figure while some are comparatively weaker (+N+C in particular). For the Materials and methods section, I question the accuracy of a couple statements: it is stated that you have NiCl in the purification buffers. Ni is divalent, so this should be more accurately reported as NiCl2. In the section on non-glycosylated proteins, the authors state that SHuffle cells are grown at 37C, but these cells are typically grown at 30C due to their thermolability (if this is correct, that is fine, but it diverges from the manufacturer's protocol). Lastly, please define for Equation 1 what b and s are.

*Reviewer #3 (Recommendations for the authors):*

The authors examined the effect of N- and C-linkers of the von Willebrand factor (VWF) A1 domain on binding to platelet glycoprotein Ibalpha (GPIbalpha). They made recombinant A1 proteins with and without the linkers and with and without O-glycans. They measured the interaction of the proteins by biolayer interferometry (BLI), isothermal titration calorimetry (ITC). Additionally, they measured the thermal and urea-induced denaturation and the hydrogen deuterium exchange (HDX) kinetics of the proteins. They found that both the N-and C-linkers lower A1 affinity, the effects of which are increased by O-glycosylation. The N-linker, but not the C-linker, stabilized the A1 domain and affected HDX exchange. They conclude that the A1 domain is regulated allosterically by the N-linker.

Overall, the conclusions of the study appear to be supported by carefully conceived and conducted experiments. The manuscript is well-written and nicely illustrated. The dissociation constants measured by BLI differ significantly from those measured by ITC, indicating systematic errors in one or both of the methods. However, the linker and glycan effects are qualitatively the same. Thus, the authors' conclusions are supported by the results of two independent physical methods.

The findings support and extend previously published findings by several groups that N- and C-linkers of the A1 domain regulate its interaction with GPIbalpha. However, the authors choose to interpret their results as being inconsistent with the proposal that N-linker and C-linker associate with one another to form an autoinhibitory module because they find that the N-linker alone is sufficient to inhibit binding of A1 to GPIbalpha. However, it is possible that the C-linker also contributes to the regulation of A1 binding to GPIbalpha, which, in fact, is supported by the authors' own the BLI and ITC results.

It would be useful for the authors to define the A1 domain in terms of the sequence that delimits it and the sequences that define the N- and C- linkers. Is the A1 delimited by residues 1272 and 1458 or by residues 1264 – 1469 (Figure 1C)?

The organization at the end of the Introduction perhaps could be improved. The last paragraph begins with "Here, to understand … we have made fundamental measurements …" preparing the reader for what the authors did. However, the authors proceed in the next few sentences to continue to cite observations from the literature.

The authors examined the effect of N- and C-linkers of the von Willebrand factor (VWF) A1 domain on binding to platelet glycoprotein Ibalpha (GPIbalpha). They made recombinant A1 proteins with and without the linkers and with and without O-glycans. They measured the interaction of the proteins by biolayer interferometry (BLI), isothermal titration calorimetry (ITC). Additionally, they measured the thermal and urea-induced denaturation and the hydrogen deuterium exchange (HDX) kinetics of the proteins. They found that both the N-and C-linkers lower A1 affinity, the effects of which are increased by O-glycosylation. The N-linker, but not the C-linker, stabilized the A1 domain and affected HDX exchange. They conclude that the A1 domain is regulated allosterically by the N-linker.

Overall, the conclusions of the study appear to be supported by carefully conceived and conducted experiments. The manuscript is well-written and nicely illustrated. The dissociation constants measured by BLI differ significantly from those measured by ITC, indicating systematic errors in one or both of the methods. However, the linker and glycan effects are qualitatively the same. Thus, the authors' conclusions are supported by the results of two independent physical methods.

The findings support and extend previously published findings by several groups that N- and C-linkers of the A1 domain regulate its interaction with GPIbalpha. However, the authors choose to interpret their results as being inconsistent with the proposal that N-linker and C-linker associate with one another to form an autoinhibitory module because they find that the N-linker alone is sufficient to inhibit binding of A1 to GPIbalpha. However, it is possible that the C-linker also contributes to the regulation of A1 binding to GPIbalpha, which, in fact, is supported by the authors' own the BLI and ITC results.

It would be useful for the authors to define the A1 domain in terms of the sequence that delimits it and the sequences that define the N- and C- linkers. Is the A1 delimited by residues 1272 and 1458 or by residues 1264 – 1469 (Figure 1C)?

The organization at the end of the Introduction perhaps could be improved. The last paragraph begins with "Here, to understand … we have made fundamental measurements …" preparing the reader for what the authors did. However, the authors proceed in the next few sentences to continue to cite observations from the literature.

---

## [Author Response]

Essential revisions:In this paper, the authors use several biophysical tools, including biolayer interferometry, isothermal calorimetry, protein denaturation, and hydrogen-deuterium exchange mass spectrometry, to systemically dissect the influence of the N- and C-terminal polypeptide linkers and O-glycan moieties of the VWF A1 domain on A1's activation state and binding affinity to platelet receptor GPIbα.Through a series of elegant experiments, the authors concluded that the N-terminal linker together with its O-linked glycans stabilize the A1 in an inactive, native conformation. Removal of the N-linker or its O-linked glycans destabilizes the A1 structure and may activate the A1 into an intermediate, higher affinity state for GPIbα binding. The authors also concluded that the C-terminal linker and its O-linked glycans do not contribute to stabilizing the A1 structure, though they prevent GPIbα binding through steric hindrance and electrostatic repulsion. The data and the figure presented in the manuscript are overall clear, and the conclusions are consistently supported by the data from multiple experimental techniques. The reported results are highly interesting for the hematology and biophysics communities.1) Define the A1 domain in terms of the sequence that delimits it and the sequences that define the N- and C- linkers and provide justification for the selection.

The justification is summarized in Results in lines 107-110: "The N and C-terminal boundaries of the constructs were based on allowing comparability to previous studies; references to the previous literature and a structural rationale for defining the minimal size of A1 as residues 1270-1463 and the maximal size of A1 plus linkers as residues 1240 to 1494 are described in Methods.” A new section in Methods, "Boundaries of A1 constructs" appears on lines 341-357.

2) Show the standard deviation of the fit instead of showing the average of two experiments in figure 2 and provide additional description for the methods used to purify protein constructs in the gel images as the bands appear differentially pure, in different amounts in figure 1.

Global non-linear least square fitting the traces at all analyte concentrations, we obtained standard fitting errors ranging from 1% to 5% of fit values. These fitting errors were smaller than the difference from the mean of 2 independent experiments shown in Figure 2; therefore, we have reported these results in the Figure 2 legend.

The proteins may have varied from 90 to 99% pure but that would not affect the measured quantities. The proteins were at different concentrations and were not adjusted to identical amounts for SDS-PAGE, which is explained in the legend. More details are given in Methods on purification and a summary has been added to the Results.

Reviewer #1 (Recommendations for the authors):In this paper, the authors use several biophysical tools, including biolayer interferometry, isothermal calorimetry, protein denaturation, and hydrogen-deuterium exchange mass spectrometry, to systemically dissect the influence of the N- and C-terminal polypeptide linkers and O-glycan moieties of the VWF A1 domain on A1's activation state and binding affinity to platelet receptor GPIbα.Through a series of elegant experiments, the authors concluded that the N-terminal linker together with its O-linked glycans stabilize the A1 in an inactive, native conformation. Removal of the N-linker or its O-linked glycans destabilizes the A1 structure and may activate the A1 into an intermediate, higher affinity state for GPIbα binding. The authors also concluded that the C-terminal linker and its O-linked glycans do not contribute to stabilizing the A1 structure, though they prevent GPIbα binding through steric hindrance and electrostatic repulsion. The data and the figure presented in the manuscript are overall clear, and the conclusions are consistently supported by the data from multiple experimental techniques. The reported results are highly interesting for the hematology and biophysics communitiesI have only a few suggestions for revision and improvement, detailed below.1) The length of the long C-linker is inconsistent among the constructs. While the constructs expressed from *E. coli* end at residue 1486, the mammalian constructs end at residue 1489. No further justification was given by the authors. Moreover, since the A2 domain starts at residue 1494, why didn't the authors include residues 1490-1493?

We have explained this above.

2) In regard to Figure 2, some of the global fits are not of high quality (e.g., A1+C with 150 mM NaCl). The authors should show the standard deviation of the fit instead of showing the average of two experiments.

Please see above.

3) It would be worth discussing the difference between the intermediate state identified by the thermal/chemical denaturation experiments and the more physiological mechanically tensioned structure. The intermediate state may or may not be the same high-affinity conformation induced by tensile pulling, since heat/urea destabilizes the entire A1 structure and tensile force/tension is not expected to spread into the structures protected by the long-ranged disulfide.

This was included in the original manusucript,: "In VWF concatemers, the ΔG required for activation by mechanical tension of the high affinity state was measured as 1.9 kcal/mol {Fu, 2017 #24942}, considerably lower than the difference in energy between the I and N states measured for A1+OGly+N+C here of 4.7 kcal/mol. These differences suggest that the directional nature of energy input by mechanical tension allows for more efficient activation of A1 than urea denaturation and may induce a distinct and less disordered intermediate state."

There are many interactions between residues internal and external to the disulfide bond, including hydrogen bonds, van der Waals contacts, burial from water. Therefore, we expect that force will have a significant indirect effect on the structure and stability of regions internal to the disulfide. This is mentioned in the Discussion.

4. Lines 223-224: It would be nice to specify the exact constructs and conditions that account for the 80-fold and 10-fold differences.

Done.

Reviewer #2 (Recommendations for the authors):[…]Some perceived weaknesses are as follows:1. The authors appropriately report a gel image of their purified protein constructs. The bands on the gel appear differentially pure, in different amounts, and there is no description in the manuscript about the relative 'diffuse' nature of the bands for glycosylated protein constructs.

This information has been added to Results.

2. The study describes the rigorous treatment of their thermal and chemical denaturation curves, and they briefly address the difference between measuring protein unfolding via intrinsic tryptophan fluorescence vs. circular dichroism. None of the curves presented illustrate error bars, but they do provide a measure of error in the associated table. Their measurements in this experiment are exclusively dependent on the environment of a single tryptophan residue in the A1 domain structure; their interpretations of the unfolding curves are warranted, but not without the caveat that it is only measuring the local environment of the tryptophan and not the entire protein structure.

We have revised " Trp-1313 is in the α1-β2 loop that has HDX dynamics that are sensitive to the presence of the N-linker, locates near the GPIbα β-finger, is mutated in VWD type 2B, and appears to be a more discrete reporter of A1 conformational state than circular dichroism."

to read

"Trp-1313 is in the α1-β2 loop that has HDX dynamics that are sensitive to the presence of the N-linker, locates near the GPIbα β-finger, is mutated in VWD type 2B, and although it may report denaturation of only a portion of A1, Trp-1313 discriminates between the native, intermediate, and denatured states of A1 similarly to circular dichroism."

My recommendations for the authors concern the gel image in figure 1 and the Materials and methods section. For figure 1, the protein bands are underwhelming. The reader would have more confidence in the study if this gel demonstrated a more uniform concentration of the protein constructs used in the study. Some of the bands are quite large and dominate the figure while some are comparatively weaker (+N+C in particular).

This has been addressed as discussed above.

For the Materials and methods section, I question the accuracy of a couple statements: it is stated that you have NiCl in the purification buffers. Ni is divalent, so this should be more accurately reported as NiCl2.

Thanks. Done.

In the section on non-glycosylated proteins, the authors state that SHuffle cells are grown at 37C, but these cells are typically grown at 30C due to their thermolability (if this is correct, that is fine, but it diverges from the manufacturer's protocol).

"SHuffle, a novel *Escherichia coli* protein expression strain capable of correctly folding disulfide bonded proteins in its cytoplasm Julie Lobstein, Charlie A Emrich, Chris Jeans, Melinda Faulkner, Paul Riggs and Mehmet Berkmen" describes multiple growth temperatures. We used 30°C after induction.

Lastly, please define for Equation 1 what b and s are.

Thanks, they were defined many lines above the equation and are now defined again immediately before Equation 1.

Reviewer #3 (Recommendations for the authors):[…]It would be useful for the authors to define the A1 domain in terms of the sequence that delimits it and the sequences that define the N- and C- linkers. Is the A1 delimited by residues 1272 and 1458 or by residues 1264 – 1469 (Figure 1C)?

Sections on this have been added as described above.

The organization at the end of the Introduction perhaps could be improved. The last paragraph begins with "Here, to understand … we have made fundamental measurements …" preparing the reader for what the authors did. However, the authors proceed in the next few sentences to continue to cite observations from the literature.

Good point. We have broken this up into two paragraphs.